# Impact of Heat Waves on Hospitalisation and Mortality in Nursing Homes: A Case-Crossover Study

**DOI:** 10.3390/ijerph182010697

**Published:** 2021-10-12

**Authors:** Ine Van den Wyngaert, Katrien De Troeyer, Bert Vaes, Mahmoud Alsaiqali, Bert Van Schaeybroeck, Rafiq Hamdi, Lidia Casas Ruiz, Gijs Van Pottelbergh

**Affiliations:** 1Academic Centre for General Practice, Department of Public Health and Primary Care, University of Leuven, 3000 Leuven, Belgium; bert.vaes@kuleuven.be (B.V.); gijs.vanpottelbergh@kuleuven.be (G.V.P.); 2Centre for Environment and Health, Department of Public Health and Primary Care, University of Leuven, 3000 Leuven, Belgium; katrien.de.troeyer@telenet.be (K.D.T.); lidia.casasruiz@uantwerpen.be (L.C.R.); 3Social Epidemiology and Health Policy, Department of Epidemiology and Social Medicine, University of Antwerp, 2000 Antwerp, Belgium; mah.saikaly@gmail.com; 4Royal Meteorological Institute, 1180 Brussels, Belgium; bertvs@meteo.be (B.V.S.); rafiq.hamdi@meteo.be (R.H.); 5Department of Physics and Astronomy, Faculty of Sciences, Ghent University, 9000 Ghent, Belgium

**Keywords:** heat wave, nursing home, hospital admissions, morbidity, mortality

## Abstract

Climate change leads to more days with extremely hot temperatures. Previous analyses of heat waves have documented a short-term rise in mortality. The results on the relationship between high temperatures and hospitalisations, especially in vulnerable patients admitted to nursing homes, are inconsistent. The objective of this research was to examine the discrepancy between heat-related mortality and morbidity in nursing homes. A time-stratified case-crossover study about the impact of heat waves on mortality and hospitalisations between 1 January 2013 and 31 December 2017 was conducted in 10 nursing homes over 5 years in Flanders, Belgium. In this study, the events were deaths and hospitalisations. We selected our control days during the same month as the events and matched them by day of the week. Heat waves were the exposure. Conditional logistic regression models were applied. The associations were reported as odds ratios at lag 0, 1, 2, and 3 and their 95% confidence intervals. In the investigated time period, 3048 hospitalisations took place and 1888 residents died. The conditional logistic regression showed that odds ratios of mortality and hospitalisations during heat waves were 1.61 (95% confidence interval 1.10–2.37) and 0.96 (95% confidence interval 0.67–1.36), respectively, at lag 0. Therefore, the increase in mortality during heat waves was statistically significant, but no significant changes in hospitalisations were obtained. Our result suggests that heat waves have an adverse effect on mortality in Flemish nursing homes but have no significant effect on the number of hospitalisations.

## 1. Introduction

The global mean temperature is rising and will continue to do so due to changes in greenhouse gas emissions. Human activities are estimated to have caused approximately 1.0 °C of global warming above pre-industrial levels, with a likely range of 0.8 °C to 1.2 °C [1]. The Intergovernmental Panel on Climate Change (IPCC) reported that global temperature will continue to rise at least until the middle of the century under all emission scenarios. The global temperature will rise by 1.5 °C or 2 °C if we do not further reduce emissions of CO_2_ and other greenhouse gases in the coming decades [2]. In Belgium, the Royal Meteorological Institute (RMI) has observed an increase in the average annual temperature of 1.9 °C since 1890. In Uccle, the average annual temperature for 2020 was 2.3 °C higher than the average for the period 1961–1990 [3]. Warming has resulted in an increased frequency, intensity, and duration of heat-related events, including heat waves in most land regions [4]. Heat waves can be defined as an extended period of days with higher temperatures than normal, but the threshold may alternate depending on which temperature the population is acclimatised to. There are consequently several definitions of a heat wave, depending on the region [5]. Local differences exist in the severity of these increases, but Europe is and will be subjected to above-average temperature changes [6]. Critical thresholds relevant for ecosystems and humans are projected to exceed the global warming temperature of 2 °C and higher [2]. In Belgium, a heat wave was observed every four years on average between 1910 and 1990. Since 2015, we have counted at least one heat wave per year [3]. The IPCC recently mentioned that observed trends in European mean and extreme temperatures cannot be explained without referring to human influence. In addition to the increase in extreme temperatures, this report also described a projected increase in pluvial flooding at a global warming temperature of 1.5 °C and higher and a further increase in river flooding at 2 °C and higher in Western and Central Europe [2]. 

Heat waves were also projected to have the greatest weather-related impact on the population’s health in Europe. The extent of this impact varies by region, probably because of differences in acclimatisation, population vulnerability, the built environment, access to air conditioning, and other factors [1]. This is also valid for Belgium where the projected increase in heat waves will be felt more intensively due to strong urbanisation [7,8,9].

Human health will be affected in many ways, inducing more challenges to our health care systems [10,11]. Numerous studies have shown a short-term rise in mortality due to prolonged periods of hot weather [12,13,14,15]. The 2003 heat wave in Western Europe, for instance, was an extremely severe event that led to more than 70,000 heat-attributable deaths, with 15,000 excess deaths in France alone [16]. The Chicago 1995 heat wave registered an average of 241 excess deaths per day, strongly exceeding the average annual of the United States heat-related deaths in a single day [17]. The elderly, children, people who are unable to care for themselves, mentally-ill people, and others with pre-existing illnesses were identified as being more vulnerable to the effects of heat [12,18,19,20]. Physiological ageing processes, potential side effects of medication, and limited mobility are factors that contribute to the higher risk of aged people [19,21,22]. 

The existence of adverse effects of heat waves on renal health and heat-related diseases, such as dehydration, heat oedema, heat cramps, heat syncope, heat exhaustion, and heat stroke, are well established. Prolonged exposure to high ambient temperatures induces a significant stress on the human cardiovascular system. Humans increase their skin blood flow and sweat rate to release heat and defend the body against increasing core temperature. These responses are necessary for thermoregulation but place a great demand on the cardiovascular system due to a large increase in cardiac output. Healthy older humans have an altered cardiovascular response to heat stress compared to younger persons. The elderly respond with diminished individual sweat gland outputs, decreased skin blood flows, reduced cardiac outputs, and smaller redistributions of blood flow from the splanchnic and renal circulations during heat stress compared to younger people [21]. Impaired possibility of evaporation leads to storage of heat, which can exacerbate the cardiovascular strain [15,21]. Even though the sweating response in the aged is often diminished, long periods of heat stress and sweating can cause a substantial reduction in plasma volume in some individuals. This loss provokes expansion of red blood cell and neutrophil counts, as well as increased plasma viscosity. Heat stress also causes the release of extra platelets into the circulation. These changes in blood properties add to increased susceptibility to cardiovascular death due to acute coronary events [21]. Yet, there is a discrepancy between findings on heat-related mortality and morbidity. More specifically, while mortality has been shown to increase, the published results on morbidity were inconsistent [11]. Less research on heat-related mortality and morbidity has been conducted in the elderly population.

In order to design effective heat adaptation strategies, it is crucial to understand the discrepancy between morbidity and mortality, as the future healthcare costs of extreme heat events are likely to increase, particularly in more heat-prone areas without effective heat adaptation adjustment. Studies have suggested that the subgroups responsible for greater heat-related costs include females, the elderly (≥64 years), low-income families, and some ethnic minorities [23].

The aim of this study is to examine the heat-related mortality and morbidity in elderly in Belgium. The target group of elderly was chosen given their high vulnerability with respect to heat and a focus on nursing homes allowed to follow a single population. The study aims to (i) assess the relationship between heat waves and mortality in nursing homes, and (ii) to investigate the effect of heat waves on hospitalisation, as a proxy for morbidity, in nursing homes. These institutions provide residential accommodation with health care, especially for elderly people with a mental or physical dependency.

## 2. Materials and Methods

This research concerns a retrospective analysis of daily mortality and hospitalisation between 1 January 2013 and 31 December 2017. Twenty-four nursing homes in Flanders, Belgium, were contacted, asking for their support for this study. Eleven of them agreed to participate and provided data on the observed period. One was excluded because it had its first residents in 2016 and could not offer information on the past years. As a result, ten nursing homes were included. Residents’ data obtained from the nursing homes contained date of admission to the home, periods of hospital admission, and date of death. All causes of death were included. Periods of hospitalisation contained both planned and acute hospitalisation. Furthermore, general data on age distribution and number of beds of the nursing home were requested. All collected data were anonymously encoded. Obtained data were used for invoicing in the nursing homes. Due to the General Data Protection Regulation, information was given on paper and manually converted into a digital format [24]. Approval for this research was given by the University of Leuven, Faculty of Medicine on 27 February 2018 (MP004837).

Measurements of daily maximum temperature (Tmax) and periods of heat waves in Ukkel, Belgium, were provided by the Royal Meteorological Institute of Belgium (RMI). Ukkel is situated 6 km south of the city centre of Brussels in a sub-urban environment. A heat wave is defined here as a period of minimum five consecutive days with Tmax exceeding 25 °C, out of which a minimum three days have a Tmax exceeding 30 °C. 

We used a time-stratified case-crossover design [25]. This design combines features of the matched case–control design and the crossover design where each subject serves as his/her own control. In addition, the inference is based on a comparison of exposure distribution rather than the risk of disease. Thus, known and unknown time-invariant confounders are inherently adjusted for. We defined two events: death and hospitalisation. Selection of the control days was based on previously published studies [26,27]. Control days were matched to the event days according to the following criteria. First, we took control days from the same month and year as the event days, both before and after the event, thus controlling for possible seasonality and long-term trends by design. Secondly, control days were selected from the same day of the week as the event days. We used conditional logistic regression models to evaluate the associations of mortality and hospitalisation with heat waves. A conditional logistic regression estimates the coefficient of exposure conditional on the matching strata and allows for the elimination of the constant related to each stratum [28]. We used separate models for four different single-day lags: the day of the event (lag 0) and up to three days before the event (lags 1, 2, and 3). We chose the lag length aforehand based on theory and the literature [11]. We estimated the odds ratios (OR) and their 95% confidence intervals (CI). All statistical analyses were conducted with RStudio (version 3.4.2) software (developed by RStudio, Boston, MA, USA) [29]. Casecross from the season package was used [30].

## 3. Results

The mean age of residents per residence exceeded 80 years on 31 December 2017 (Table 1). The number of residents differed between nursing homes (Table 1). From 1 January 2013 until 31 December 2017, 1888 residents living in the 10 included nursing homes died. For the reference period, 3048 hospital admissions were registered. The average number of deaths and hospitalisations calculated per 1000 residents is shown in Table 1. The variance of hospitalisation and number of deaths is represented in Table 1 as well.

Heat waves occurred each year, except in 2014. There were 23 heat wave days during the reference period. The longest heat waves date from 2013 and 2015. Heat conditions lasted for seven days in both periods (Table 2). The average number of hospitalisations remained stable throughout the years. The mortality numbers increased during the five investigated years (Table 2).

Mortality was directly associated with heat waves. This association was significant on lags 0, 2, and 3. For hospitalisation, the associations were weaker and not statistically significant. The OR of the heatwave effect on mortality and hospitalisation is shown along with 95% CI at lag 0, 1, 2, and 3 (Table 3). 

## 4. Discussion

This work focused on the health consequences of heat for elderly, living in nursing homes. We found a statistically significant effect of heat waves on mortality but no statistically significant effect on hospitalisation.

No important differences in the annual number of hospitalisations between years with and without a heat wave could be found, as expected. The consequences of heat must be rather extensive to affect the average annual mortality or the number of hospitalisations. 

An increase in mortality due to heat waves has been described in numerous studies [10,13,14,15,22,23,24,25,26,27,28,29,30,31,32,33,34]. A recent research showed Eastern Europe had the highest heat-related excess death rate [35]. Time-series analyses results proved that higher diurnal temperature range was associated with excess mortality [36]. Fouillet et al. conducted an analysis of the August 2003 heat wave in France. They found a relative increase in mortality of 91% in nursing homes [31]. Borst et al. reported an increase of 50% in mortality rate in Dutch nursing homes, when the maximal weekly temperature exceeded 25 °C [32]. Our results are in line with this finding. Baccini et al. identified a stronger association between heat and mortality in people aged above 74 years than earlier in life [34]. Various studies described cardiovascular mechanisms as underlying causes of the excess of deaths [10,21,33,37]. Several studies examined the relation between heat and, respectively, respiratory and cerebrovascular mortality. However, these researches yielded conflicting results [10,33,34,35,36,37].

Our analyses did not find a significant relation between heat waves and hospital admissions for the elderly. Inconsistent results were found among studies examining the impact of heat waves on the hospitalisation rate. Some authors reported an increase in hospital admissions for adults as well as for the elderly due to heat exposure [15,19,38]. Wu et al. postulated that temperature variability was significantly associated with asthma hospitalisation in Brazil [39]. Wei et al. found that higher temperature fluctuations were associated with an increased risk of dementia hospitalisations [40]. In addition, Liss et al. documented in 2017 an enlarged effect of the first seasonal heat wave on heat-related hospitalisation in older adults. In addition, they suggested to focus on the first heat wave to intensify preventive measures [19]. Other studies did not show a significant rise during heat waves as described by Kovats during the UK heat wave of 1995 [13]. Among studies that explored morbidity, there was consensus on the unfavourable health effects of heat waves on heat-related and renal diseases [11,14,31,38]. Most of the studies postulated the absence of impact of heat exposure on cardiovascular morbidity [10,31]. Bunker et al. described a heat-related association with respiratory disease among the elderly as confirmed in other studies [13,14,19]. However, the meta-analysis showed ambivalent results for cardiovascular and respiratory morbidity [11,41].

The impact of hot weather on mortality in nursing homes was not proportional to the effect on hospital admissions. Because of these contrasting patterns, the question was raised as to why the number of deaths increased with increased temperature. Some studies postulated a knowledge gap hypothesis: During heat waves, many deaths occur rapidly before patients could reach the hospital [13,14,33]. According to some authors, the presence of an increase in cardiovascular mortality and the absence of an effect of heat on cardiovascular admissions suggest a sudden death due to circulatory disease [13,33]. 

Further research is needed to confirm this hypothesis in nursing homes. Another possible explanation can be found in the fact that in nursing homes, the policy is rather to keep the person in the home instead of hospitalising, prioritising the person’s well-being over curation. During Advance Care Planning, the wishes and needs of patients and their relatives are discussed with a health care worker and recorded in their medical records. Some elderly noted their will to quit hospitalisation. Moreover, this study concerns a frail population. Their death was possibly only slightly accelerated by the heat wave [31].

Although there was a discrepancy between heat-related mortality and morbidity, we assume there is a sufficient avoidable burden to evaluate preventive measures during extreme heat events. In response to the heat wave in 2003, action plans were drawn up by several countries to manage such a crisis situation. Since 2005, Belgium has a Heat Health Warning System (HHWS). This system considers minimum and maximum temperature values and ozone values to trigger heat warnings. The intention of the HHWS is to reduce the health impact of heat on vulnerable persons [6]. HHWS, when accompanied by specific health interventions, are considered to be effective in reducing deaths during a heatwave [15]. Because our results suggest that there is still a significant effect of heat on mortality in nursing homes in Flanders, we recommend a process evaluation of the Belgian HHWS in nursing homes to further prevent deaths due to extreme heat events. 

The major strength of this analysis is the completeness of the data since all residences were included in the five-year observational period. In addition, both mortality and morbidity were investigated. Furthermore, we decided to focus on the elderly, especially those living in nursing homes. Less research on mortality and morbidity has been conducted in this population. This population is particularly valuable to study the discrepancy between morbidity and mortality because aged individuals are more vulnerable to the cardiovascular effects of heat than younger adults, as we previously described. In addition, due to deterioration of brain function, the sensibility of thirst is decreased. Insufficient physical fitness, the presence of chronic underlying disease, polypharmacy, social isolation, financial problems, and dependency on others contribute to the higher risk of heat-related diseases [18,21]. Subsequently, we expect to detect an effect of heat waves earlier in this high-risk group. So, it is useful to investigate mortality in nursing homes. Furthermore, if we can prevent heat-related diseases in elderly, more effects of prevention on health care expenses and quality could be expected [23].

Some limitations to this study should be noted. First, we did not adjust for the concentration of outdoor air pollutants in this research. A period of extreme heat involves increased radiation, little wind, and very little rain. As a result, production of pollutants, such as ozone, occurs in the atmosphere near the Earth’s surface. In the meantime, higher concentrations of ozone and fine dust amplify the effect of extreme heat [12]. However, some studies suggest that the heat effects are likely to continue after control for air pollution [4,42]. Finally, a rather small number of nursing homes was included in this research. Information about gender differences, co-morbidities, and diagnosis of death was not investigated.

The differences in the effect of heat on mortality and morbidity have been described in various studies. Different factors contributed to the discrepancies. First, there is no clear-cut definition of a heat wave in Europe [4,20,33]. Furthermore, differences in the intensity of exposure, separate susceptibility of the examined populations, and variation in health and social care services direct to heterogeneity [13,20]. Methodology and choice of outcome are also accounted for [11,20]. Another point of discussion is the exposure indicator. In this research, Tmax was chosen to define a heat wave. Previous studies considered a range of different exposure variables [10,42]. Some authors suggested apparent temperature to be a better measure than temperature alone. This index incorporates temperature, humidity, and sometimes wind speed, which could indicate relative human discomfort [10]. However, in some assessments, apparent temperature was not a superior predictor of mortality to mean temperature [15,25].

## 5. Conclusions

This study suggests that heat waves have an adverse effect on mortality in nursing homes while no significant effects could be found in hospitalisations. Our attention must be directed to high-risk groups of heat, including the elderly. Anticipating heat waves, a process evaluation of the Belgian HHWS in nursing homes and preventing healthcare costs due to extreme heat events must be important goals for the future.

## Figures and Tables

**Table 1 ijerph-18-10697-t001:** Description of the 10 participating nursing homes including the average number of daily hospital admissions and deaths calculated per 1000 residents per nursing home, across five years of reference period. Variance (Var), hospitalisation (hosp).

Nursing Home	Province	Number of Beds	Average Age	Hospitalisations	Var(Hosp)	Deaths	Var(Deaths)
1	Flemish Brabant	>200	80y	1.03	0.01	0.60	0.03
2	Antwerp	50–100	86y	1.90	0.11	1.21	0.36
3	Antwerp	151–200	86y	1.14	0.03	0.77	0.12
4	Antwerp	50–100	87y	1.15	0.09	0.83	0.01
5	Antwerp	50–100	85y	1.77	0.04	0.93	0.32
6	Antwerp	50–100	86y	1.18	0.07	0.82	0.14
7	Antwerp	151–200	84y	1.11	0.04	0.68	0.04
8	Antwerp	50–100	85y	0.93	0.04	0.64	0.14
9	Antwerp	101–150	86y	0.87	0.03	0.57	0.06
10	Flemish Brabant	101–150	88y	1.14	0.07	0.62	0.06

**Table 2 ijerph-18-10697-t002:** Average number of hospitalisations and deaths calculated per 1,000 residents per day. Variance (Var), hospitalisation (hosp).

Year	Heat Wave	Number of Days	Hospitalisations	Var(Hosp)	Deaths	Var(Deaths)
2013	21/7–27/7	7	1.18	0.14	0.42	0.02
2014	–	0	1.17	0.08	0.64	0.05
2015	30/6–5/7	7	1.29	0.24	0.82	0.07
2016	23/8–27/8	5	1.25	0.18	0.90	0.20
2017	18/6–22/6	5	1.21	0.15	1.06	0.14

**Table 3 ijerph-18-10697-t003:** Odds ratios (OR) at lag 0, 1, 2 and 3 days for hospitalisations (A) and deaths (B). 95% confidence interval (95% CI). Bold indicates the significant results at lag 0, 2, and 3.

**A**	**Hospitalisation**	**Lag**	**OR**	**95% CI**
		0	0.96	0.67–1.36
		1	0.98	0.69–1.39
		2	1.14	0.82–1.59
		3	1.14	0.81–1.61
**B**	**Deaths**	**Lag**	**OR**	**95% CI**
		0	**1.61**	1.10–2.37
		1	1.29	0.84–1.97
		2	**1.62**	1.06–2.50
		3	**1.71**	1.15–2.56

## Data Availability

The data presented in this study are available on request from the corresponding author. The data are not publicly available due to ethical restrictions.

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
