# Peer review of "Impact of Heat Waves on Hospitalisation and Mortality in Nursing Homes: A Case-Crossover Study"

_ijerph, 2021, doi:10.3390/ijerph182010697_

Round 1

Reviewer 1 Report

This paper is a useful addition to the literature. There are few instances where the author needs to make some improvements. Below are my comments:

- line 43: more recent temperature records (2019, 2020, 2021) should be mentioned not only for Belgium but for whole of Europe.

- lines 49 - 59: more details about how temperature can lead to death are missing.

- The Introduction should highlight the relevance of the topic, the novelty of the results, the sample's choice, the method's appropriateness, and the contribution to the literature.

- Literature review is partial and incomplete, and some recent and relevant contributions should be cited and discussed (i.e., https://doi.org/10.1016/S2542-5196(21)00081-4; https://doi.org/10.1016/S2542-5196(20)30222-9; http://dx.doi.org/10.1136/thoraxjnl-2020-216549; https://doi.org/10.1016/j.envint.2018.12.054; https://doi.org/10.1088/1748-9326/ab1cdb)

Methods:

- the selection of control days should be justified based on other previously published studies. In addition, these data can be better described from a table.

- the choice of the model should be justified based on other previously published studies. 

- Describe how conditional logistic regression model works, and how the odds ratio was determined, including their equations.

- lag analysis should be justified based on other previously published studies.

- line 112: the used packages should be mentioned. Also inform if the codes are available on github.

Results:

- why not test more lags? There are specific estimates for selecting optimal lag lengths, such as AIC.

Conclusions:

- Conclusions should be better presented.

Author Response

- line 43: more recent temperature records (2019, 2020, 2021) should be mentioned not only for Belgium but for whole of Europe.

* Thank you for the suggestion. These records were added in the study.

- lines 49 - 59: more details about how temperature can lead to death are missing.

* We thank the reviewer for the suggestion. Details on how temperature can lead to death were transferred from the discussion to the introduction.

- The Introduction should highlight the relevance of the topic, the novelty of the results, the sample's choice, the method's appropriateness, and the contribution to the literature.

* Thank you for noticing, these parts were described in the manuscript. We provide an overview where these parts can be found.

  • The relevance of the topic: Line 35-55
    We discussed global warming and the associated increase in the amount of heat waves and their impact on human health.
  • The sample’s choice: Line 96-97
    The target group of elderly was chosen given their high vulnerability with respect to heat and a focus on nursing homes allowed to follow a single population
  • The novelty of the results: Line 87
    Less research on heat-related mortality and morbidity has been done in the elderly population.
  • The method’s appropriateness: Thank you for the suggestion. We prefer to describe this part under the method section.
  • The contribution to the literature: Line 98-100
    We investigated the relationship between heat waves and mortality in nursing homes, and the effect of heat waves on hospitalisation, as a proxy for morbidity, in nursing homes.

- Literature review is partial and incomplete, and some recent and relevant contributions should be cited and discussed (i.e., https://doi.org/10.1016/S2542-5196(21)00081-4; https://doi.org/10.1016/S2542-5196(20)30222-9; http://dx.doi.org/10.1136/thoraxjnl-2020-216549; https://doi.org/10.1016/j.envint.2018.12.054; https://doi.org/10.1088/1748-9326/ab1cdb)

* Thank you for these suggestions. We included these references in the study.

Methods:

- the selection of control days should be justified based on other previously published studies. In addition, these data can be better described from a table.

* We agree with the reviewer that a reference should be provided, justifying the selection of control days. We used the time stratified method to control for seasonal and long-term trends by design (reference Janes 2005: DOI: 10.1002/sim.1889).

* As discussed in our paper, we chose to select our referent days in the same day of the week, the same month and year as the index day. Stratification for day of the week was performed, since it is known that mortality is associated with the day of the week (DOI: 10.1007/s00134-004-2170-3).

* However, we do not understand how the selection of control days could be described in a table. We are open to any further discussion on this matter if the reviewer could clarify this question.

- the choice of the model should be justified based on other previously published studies.

* In our opinion, the choice of the statistical model should be based on our research questions, the occurrence relation and the type of data. Our outcome is binary, so we chose a logistic regression. In addition, because the case days are matched to the control days, and so the observations within each stratum are not independent by default, a conditional logistic regression is in our opinion appropriate.

- Describe how conditional logistic regression model works, and how the odds ratio was determined, including their equations.

* A conditional logistic regression estimates the coefficient of exposure conditional on the matching strata and allows for the elimination of the constant related to each stratum. We are interested mainly in the coefficient of the independent variable rather than the different stratum as discussed in the paper of Mostofsky et al (2018) (https://www.ncbi.nlm.nih.gov/pmc/articles/PMC6167149/).

- lag analysis should be justified based on other previously published studies.

* We can justify that for high temperature, the effect is more immediate compared with cold wheather. There is an acute effect of heat on mortality and hospitalisation, but we know from literature that this effect also lasts for the next few days. We can include previous studies that showed the effect in these lag periods (https://www.ncbi.nlm.nih.gov/pmc/articles/PMC4454966/).  

- line 112: the used packages should be mentioned. Also inform if the codes are available on github.

* Used package is casecross from season package (https://rdrr.io/rforge/season/man/casecross.html). We used a fairly easy code; variables were plugged in the order.

Results:

- why not test more lags? There are specific estimates for selecting optimal lag lengths, such as AIC.

* We thank the reviewer for the suggestion to include more lags in the analysis. However, our aim was to estimate the immediate effect of heat waves on mortality/hospitalizations in the elderly. To estimate delayed effects of heat waves, the case-crossover design is inappropriate. In addition, the effect of heat on mortality is known to be immediate (bvb DOI: 10.1097/EDE.0b013e318190ee08). We chose the lag length aforehand based on theory and the literature and consider this strategy more appropriate than basing our choice on model fit criteria such as AIC.

* We however acknowledge that this strategy was not clearly explained. Therefore, we now added a sentence in the methods section.

Conclusions:

- Conclusions should be better presented.

Thank you for this comment. We made the conclusion more clear.

Reviewer 2 Report

I have reviewed the manuscript on impacts of heat waves on mortality in elderly citizens in Belgium with keen interest. I have some comments for the authors which may help improve the manuscript.

  • Firstly, as this study involves humans and confidential information about their health, it is not clear in the manuscript if the eleven nursing homes that agreed to participate dealt with ethical issues regarding patients information.  
  • I feel the introduction needs to be grown a bit. For example the trends and projections of rapidly rising temperatures need to refer to the latest IPCC AR6 2021 report with focus on the region (Europe) on the study
  • On reading further, I realized a lot of what I expected in the Introduction is provided in the discussion. I was hoping the authors could indicate that there are several definitions of heat waves in different regions and that while in this study 25°C may be considered hot, in many tropical regions it is comfortable temperature. How heat waves affect the human body through circulation, blood pressure, heart rate also needs to be in the introduction, so there is a clear understanding of heat waves and also how they affect human health, and also why focus on the elderly (vulnerabilities).
  • The authors are silent about the elderly in their sample who have had a long history of health problems and those who may be considered healthy. I am thinking of the elderly with a history of heart disease, hypertension, cancer etc - such that a heat wave event may just be a small trigger of the inevitable.
  • Thus the conclusion that "mortality was directly associated with heat waves"could be misleading. Statistical models do not necessarily mean causation.

Author Response

  • Firstly, as this study involves humans and confidential information about their health, it is not clear in the manuscript if the eleven nursing homes that agreed to participate dealt with ethical issues regarding patients information.
    • Each Belgian research is submitted to the Ethics Committee that determines which measures are necessary with regard to data protection. This investigation was approved by the Ethics Committee without the need for further action (Line 115-116). This is after all a retrospective study in which only anonymous data were used.
  • I feel the introduction needs to be grown a bit. For example the trends and projections of rapidly rising temperatures need to refer to the latest IPCC AR6 2021 report with focus on the region (Europe) on the study
    • Thank you for the suggestion. These trends were added in the manuscript.
  • On reading further, I realized a lot of what I expected in the Introduction is provided in the discussion. I was hoping the authors could indicate that there are several definitions of heat waves in different regions and that while in this study 25°C may be considered hot, in many tropical regions it is comfortable temperature. How heat waves affect the human body through circulation, blood pressure, heart rate also needs to be in the introduction, so there is a clear understanding of heat waves and also how they affect human health, and also why focus on the elderly (vulnerabilities).
    • We thank the reviewer for the suggestion. Parts of the discussion were transferred to the introduction.
  • The authors are silent about the elderly in their sample who have had a long history of health problems and those who may be considered healthy. I am thinking of the elderly with a history of heart disease, hypertension, cancer etc - such that a heat wave event may just be a small trigger of the inevitable.
    • Thank you for this comment. We fully agree that this must be added to the manuscript.
  • Thus the conclusion that "mortality was directly associated with heat waves"could be misleading. Statistical models do not necessarily mean causation.
    • Thank you fort this comment. When we use the term ‘association’, we refer to the general relationship between variables, which does not imply causation.

Reviewer 3 Report

The methods and findings of this paper are set out clearly and concisely, and the explanation of the findings in the Discussion section is plausible.

Much of the discussion on mortality is about mechanisms of cardiovascular deaths in subjects with pre-existing heart disease, yet as the authors point out, excess cardiovascular mortality has not been a consistent finding in published research.  As this study specifically addresses the effects on nursing home residents it would be helpful to know whether there actually was an excess of cardiovascular deaths in this study population.  With 1888 deaths overall, would there not be enough CVD deaths to generate an OR?

Two mechanisms are proposed for increased CVD mortality – the first being reduced skin perfusion leading to a lower sweat rate leading to heat storage, and the second excess sweat loss leading to increased blood viscosity.  These appear to be contradictory.  Are you saying that these two mechanisms occur simultaneously? This is a critical issue since the latter contingency implies some dehydration, while the first suggests that rehydration of subjects with heart failure may lead to overload.

The definition of heat waves as 3 consecutive maxima of 250C is a surprise to this reviewer in Australia where a heat wave is defined as five days of 350C or more, or three days of 400C or more. The authors have pointed out the limitations of daily maximum temperature (Tmax) as an appropriate exposure metric. The ideal would be an index accounting for temperature, absolute humidity and air velocity (the last of these being a critical factor in heat stress indoors), recorded within the nursing home (ie not ambient).  The fact that this study showed excess mortality despite the use of a relatively crude exposure index is noteworthy, and suggests that these nursing homes need to upgrade their air-conditioning systems.

Line 141 (heading of Table 3: “ratios” not “ratio’s”

Line 207: I think “Heath” should be “Heat”

Author Response

  • Much of the discussion on mortality is about mechanisms of cardiovascular deaths in subjects with pre-existing heart disease, yet as the authors point out, excess cardiovascular mortality has not been a consistent finding in published research.  As this study specifically addresses the effects on nursing home residents it would be helpful to know whether there actually was an excess of cardiovascular deaths in this study population.  
    • At present we do not have data on the cause of death in the participating patients (only the date of death is available), because these data are privacy sensitive. In addition, the exact cause of death in the elderly population is not always known, since few autopsies are performed in these patients. It is therefore not possible to look at the number of cardiovascular deaths in this study population.
  • Two mechanisms are proposed for increased CVD mortality – the first being reduced skin perfusion leading to a lower sweat rate leading to heat storage, and the second excess sweat loss leading to increased blood viscosity.  These appear to be contradictory.  Are you saying that these two mechanisms occur simultaneously? This is a critical issue since the latter contingency implies some dehydration, while the first suggests that rehydration of subjects with heart failure may leadto overload.
    • Thank you for this comment, this was indeed unclear in the manuscript. We have corrected this now in the manuscript. These processes do not occur simultaneously. In most elderly people there is a decreased sweat output during heat exposure. Another process involves increased viscosity, mainly occurring when there is prolonged heat stress.
  • The definition of heat waves as 3 consecutive maxima of 250C is a surprise to this reviewer in Australia where a heat wave is defined as five days of 350C or more, or three days of 400C or more. The authors have pointed out the limitations of daily maximum temperature (Tmax) as an appropriate exposure metric. The ideal would be an index accounting for temperature, absolute humidity and air velocity (the last of these being a critical factor in heat stress indoors), recorded withinthe nursing home (ie not ambient).  The fact that this study showed excess mortality despite the use of a relatively crude exposure index is noteworthy, and suggests that these nursing homes need to upgrade their air-conditioning systems.
    • Thank you for this comment. We understand why our definition of a heat wave is a surprise for an Australian reviewer. As mentioned in the discussion (and now also added to the introduction), each region has its own definition of a heat wave. It is adapted to the region and depends on the temperatures to which the population is acclimatised to. We used the official Belgian definition of a heat wave.
    • We agree with the comment that Belgian nursing homes need to upgrade their air-conditioning systems, since air-conditioning is not standard provided in nursing homes. Moreover, the buildings in Beldium are designed to retain heat rather than losing it.
  • Line 141 (heading of Table 3: “ratios” not “ratio’s”
    • Thank you for this comment, this was indeed a mistake. We have corrected this now in the manuscript.
  • Line 207: I think “Heath” should be “Heat”
    • Thank you for this comment, this was indeed a mistake. We have corrected this now in the manuscript.

Round 2

Reviewer 1 Report

The suggested changes were performed or justified satisfactorily.

Reviewer 2 Report

The authors  have effected the recommendations adequately